# The Influence of Thyroid Pathology on Osteoporosis and Fracture Risk: A Review

**DOI:** 10.3390/diagnostics10030149

**Published:** 2020-03-07

**Authors:** Dragos Apostu, Ondine Lucaciu, Daniel Oltean-Dan, Alexandru-Dorin Mureșan, Cristina Moisescu-Pop, Andrei Maxim, Horea Benea

**Affiliations:** 1Department of Orthopedics and Traumatology, University of Medicine and Pharmacy “Iuliu Hatieganu”, Cluj-Napoca 400132, Romania; apostudragos@yahoo.com (D.A.); olteandandaniel@yahoo.com (D.O.-D.); andrei.v.maxim@gmail.com (A.M.); beneahorea@yahoo.com (H.B.); 2Department of Oral Health, University of Medicine and Pharmacy “Iuliu Hatieganu”, Cluj-Napoca 400012, Romania; 3Department of Sport and Physical Education Science, University of Babes-Bolyai, Cluj-Napoca 400376, Romania; alex.d.muresan@gmail.com; 4Nuclear Medicine Department, “Prof. Dr. Ioan Chiricuța” Institute of Oncology, Cluj-Napoca 400015, Romania; popcrisro@yahoo.com

**Keywords:** Thyroid, bone mineral density, osteoporosis, fracture, hypothyroidism, subclinical hypothyroidism, subclinical hyperthyroidism

## Abstract

Thyroid hormones are important factors that regulate metabolism and cell differentiation throughout the human body. A complication of thyroid pathology is represented by an alteration of the bone metabolism which can lead to osteoporosis and fragility fractures, known to have a high mortality rate. Although there is a consensus on the negative impact of hyperthyroidism on bone metabolism, when referring to hypothyroidism, subclinical hypothyroidism, or subclinical hyperthyroidism, there is no general agreement. The aim of our review was to update clinicians and researchers about the current data regarding the bone health in hypothyroidism, subclinical hypothyroidism, and subclinical hyperthyroidism patients. Thyroid disorders have an important impact on bone metabolism and fracture risk, such that hyperthyroidism, hypothyroidism, and subclinical hyperthyroidism are associated with a decreased bone mineral density (BMD) and increased risk of fracture. Subclinical hypothyroidism, on the other hand, is not associated with osteoporosis or fragility fractures, and subclinical hyperthyroidism treatment with radioiodine could improve bone health.

## 1. Introduction

Thyroid hormones are important factors which regulate metabolism and cell differentiation throughout the human body. Their receptors are localized in the nervous system, pituitary gland, lungs, heart, liver, muscle, bones, testis, placenta, and other tissues. Therefore, their increase or decrease has an impact on the whole body. These changes in thyroid hormone levels occur in hypothyroidism, hyperthyroidism, as well as subclinical hypothyroidism and subclinical hyperthyroidism. An important effect is represented by changes in bone metabolism, which frequently leads to a decreased bone mineral density called osteoporosis. This complication can lead to an increased risk of fragility fractures, usually associated with a high risk of mortality within the first year. These effects appearing on the bone tissue have been well documented in the case of hyperthyroidism. On the other hand, there is no consensus when referring to hypothyroidism, subclinical hypothyroidism, or subclinical hyperthyroidism. The aim of our review is to update clinicians and researchers about the current data regarding the bone health in hypothyroidism, subclinical hypothyroidism, and subclinical hyperthyroidism patients. Moreover, it will discuss the treatment options in such cases.

## 2. Materials and Methods

We have searched the PubMed database for the following keywords: “hypothyroidism osteoporosis”, “hypothyroidism fracture”, “hypothyroidism bone”, “subclinical hypothyroidism osteoporosis”, “subclinical hypothyroidism fracture”, “subclinical hypothyroidism bone”, “subclinical hyperthyroidism osteoporosis”, “subclinical hyperthyroidism fracture”, “subclinical hyperthyroidism bone”, ‘’autoimmune thyroid bone metabolism”, ‘’autoimmune thyroid bone mineral density”, ‘’autoimmune thyroid osteoporosis”, ‘’autoimmune thyroid fracture”, “Hashimoto osteoporosis”, and ‘’Hashimoto fracture”. We have included clinical trials and meta-analyses from 2002 until 2020. The bone studies reviewed in our study included dual-energy X-ray absorptiometry (DXA) scan, bone turnover markers and Fracture Risk Assessment Tool (FRAX) score. From a total of 3505 articles, only 41 fulfilled our inclusion criteria.

## 3. Bone Health

Osteoporosis is represented by a low bone mass and it is defined as the bone mineral density of 2.5 standard deviations of that of young adults (T-score), measured using a dual-energy X-ray absorptiometry (DEXA) [1]. It is the result of an imbalance between the processes of bone resorption and bone formation. It has a high incidence in the general population, such that up to 50% of post-menopausal women are affected by osteoporosis [2]. Osteoporosis can be either primary or secondary. For primary, the causes are age and menopause [3]. Secondary causes are numerous and they include liver disease, renal failure, rheumatoid arthritis, chronic obstructive pulmonary disease, endocrine disorders, gastrointestinal disorders, hematologic disorders, nutrition disorders, central nervous system disorders, human immunodeficiency virus infection, systemic lupus erythematosus, and diverse medications [3,4,5]. In the United States, women over 65 years old or those with an increased fracture risk using FRAX score are recommended to be screened for osteoporosis [3]. Clinically, osteoporosis has no symptoms, but it can lead to loss of height, back pain, change in posture or intense pain following a fracture, as it increases the fracture risk by two-fold for every standard deviation below the mean of a young adult [2,6].

Regarding the complications of osteoporosis, the most frequent osteoporotic fractures are vertebral, where two-thirds are asymptomatic, followed by distal radius and hip fractures [6]. It is considered that 30–40% of people with osteoporosis will have an osteoporotic fracture during their lives [1]. This is important, as vertebral and hip fractures are life-threatening pathologies in the elderly population. 

Following the diagnosis of osteoporosis, patients are recommended supplement intake such as calcium at a daily dose of 1000 mg for men aged 50–70 years and 1200 mg for men aged over 70 years and women over 50 years old [1]. Moreover, vitamin D intake is also needed at a dose of 600 IU/day until 70 years old and 800 IU/day afterward [1]. Preferably, the quantity of vitamin D should be adjusted according to the serum 25 (OH) D levels [1]. Other management strategies for osteoporotic patients in order to reduce the fracture risk include the decrease of alcohol consumption, less than two servings of caffeine drinks daily, beginning a regular physical exercise program, and fall-prevention programs.

## 4. Physiopathology of Thyroid’s Effect on Bone Metabolism

The bone remodeling cycle is a key element for the bone changes found in thyroid pathology. The bone undergoes a continuous process of bone formation and bone resorption throughout the lifetime, called the bone remodeling cycle [7]. The bone remodeling process is started by osteoclasts, which are osteoblast-derived cells. They are connected to each other by a dendritic network and initiate the bone resorption performed by osteoclasts. After the osteoclasts finish the osteolytic process, osteoblasts perform bone formation at the site. Apart from local factors, the bone remodeling process is regulated by systemic factors such as calcitonin, parathyroid hormone, vitamin D3, estrogen, thyroid hormone, glucocorticoids, and growth hormones [8]. 

The T3 hormone acts on TRα receptors on both osteoblasts and osteoclasts (Figure 1) [9]. On osteoblasts, it is considered to increase osteoblast formation and bone formation, while on the osteoclasts it could increase osteoclast formation and the bone resorption process (Figure 1) [9]. It is not yet clear whether T3 acts on osteoclasts directly or indirectly using the osteoprotegerin/receptor activator of nuclear factor kappa-B ligand (OPG/RANKL) pathway (Figure 1) [9]. The pituitary gland can also act directly on the bone cells by thyroid-stimulating hormone (TSH) action on the TSH receptor (TSHR) found in both osteoblasts and osteoclasts, in a similar manner to T3 [9,10].

The literature shows that the normal bone remodeling cycle of approximately 200 days is reduced to almost 100 days in the case of hyperthyroidism and increased to approximately 700 days in the case of hypothyroidism [11]. Some data support the idea that hyperthyroidism reduces bone mineral density by almost 10% for every bone remodeling cycle, while hypothyroidism increases bone mineral density by almost 17% for every bone remodeling cycle [11]. The lower bone mass found in hyperthyroidism results in an increased risk of fractures [10]. On the other hand, although the bone mass is considered to be increased in hypothyroidism by some studies, the fracture risk is increased due to increased stiffness of the bone [10].

Hyperthyroidism is generally accepted to decreases bone mineral density (BMD) and increases the risk of fractures. On the other hand, clinicians have different opinions on the impact of hypothyroidism, subclinical hypothyroidism, subclinical hyperthyroidism, and their treatments on bone pathology. 

Thyroid disorders can lead to osteoporosis and increase the risk of fractures. Hip fractures are an important cause of presentation in the emergency departments worldwide and are associated with an increased cost [12]. Most importantly, hip fractures cause a reduction of functional status, a decrease in mobility, and a decrease in social independence [12]. Furthermore, these fractures are associated with an increased risk of mortality, ranging between 9.7% to 34.8% in the first year following the traumatism [12]. Therefore, any effort to reduce the incidence of hip fractures by limiting the reduction of bone mineral density is important for patients with associated thyroid pathologies. 

We reviewed the PubMed database from 2002 to 2020 in search of clinical studies on the impact of hypothyroidism, subclinical hypothyroidism, subclinical hyperthyroidism, and their treatments on bone metabolism and incidence of fragility fractures (Table 1, Table 2 and Table 3).

## 5. Thyroid Pathology

Hypothyroidism is represented by thyroid hormone deficiency and it is a common pathology worldwide, with a prevalence of up to 7% in the general population [47]. The causes of hypothyroidism can be divided into primary, central, and peripheral [47]. Primary hypothyroidism is represented by thyroid hormone deficiency and includes chronic autoimmune thyroiditis as well as iodine deficiency or radioiodine treatment. Central hypothyroidism is represented by thyroid stimulating hormone (TSH) deficiency or by thyrotropin-releasing hormone (TRH) deficiency and includes pituitary tumors, pituitary dysfunction, or hypothalamic dysfunction. Peripheral hypothyroidism is represented by the peripheral resistance to thyroid hormones and includes decreased sensitivity to thyroid hormones and consumptive hypothyroidism [47].

Diagnosis of hypothyroidism requires examining the levels of thyroid-stimulating hormone (TSH) and free thyroxine. If the TSH level is above the normal range, while the free thyroxine (FT4) level is below the normal range, then primary hypothyroidism is diagnosed. When central hyperthyroidism (both secondary and tertiary hypothyroidism) is present, the TSH level can be either normal or low, while the FT4 level is low. Peripheral hypothyroidism is a congenital disorder and is not the subject of our research. 

The current treatment of hypothyroidism is represented by levothyroxine monotherapy. The dose is adjusted until TSH levels are normalized [47]. Additionally, a levothyroxine–liothyronine combination therapy can be used in some subgroups of patients [47]. 

Complications of hypothyroidism include goiter, cardiovascular disease, myxedema, infertility, and mental health issues.

Subclinical hypothyroidism is a mild form of hypothyroidism defined as a higher TSH level with a normal serum free thyroxine (FT4) level [48]. It is thought that over 20% of women over the age of 75 years present with this condition [48]. Causes of subclinical hypothyroidism include autoimmune thyroiditis, radioiodine treatment, subtotal thyroidectomy, subacute thyroiditis, drug-induced, and TSH receptor [48]. Subclinical hypothyroidism is more likely to present fatigue compared to euthyroid patients [48]. A review showed associations between subclinical hypothyroidism and coronary heart disease in younger patients with high TSH levels, cardiac failure in patients with TSH > 10 mIU/L, cerebrovascular disease in younger patients with high TSH levels, adverse lipid profiles, and type 2 diabetes [48]. No association was found between subclinical hypothyroidism and impaired cognition, frailty, and neuropsychological function [48]. There is currently no consensus on whether subclinical hypothyroidism should be treated [48].

Subclinical hyperthyroidism is the mild form of hyperthyroidism and it is defined as a low TSH level associated with normal levels of free thyroxine (FT4) and free triiodothyronine (FT3) [49]. It can be caused by Graves’ disease, multinodular toxic goiter, autonomous functioning thyroid adenoma, or an iatrogenic condition in differentiated thyroid carcinoma [49,50,51]. Subclinical hyperthyroidism increases the risk of atrial fibrillation, heart failure, and overall mortality (especially if TSH levels are <0.1 mIU/L) [49]. Treatment is recommended in patients with TSH < 0.1 mIU/L and aged 65 or older, with heart disease, hyperthyroid symptoms, osteoporosis, or post-menopausal. In the case of a TSH level between 0.1 and 0.4 mIU/L and the mentioned comorbidities, the treatment is controversial [49].

Hypothyroidism and bone health. Hypothyroidism has been shown to decrease BMD in the majority of studies. TSH values both above and below the reference values have been shown to decrease the BMD (Table 1, Table 2 and Table 3) [14,16,17,32,33]. Only one prospective study performed by Tuchendler and Bolanowski in 2013 using 119 participants did not show a negative effect on BMD in the case of hypothyroidism, except in the case of hyperthyroidism (Table 1, Table 2 and Table 3) [15]. Regarding the incidence of fracture risk in hypothyroidism patients, we found two studies using 92,341 and 16,249 patients, that showed a higher incidence of fractures in these patients (Table 1, Table 2 and Table 3) [34,35]. As a result, we believe hypothyroidism should be considered a risk factor for osteoporotic fractures.

Can the treatment of hypothyroidism reverse the negative effect on bone? Studies have shown that hypothyroidism overtreatment with levothyroxine is the main negative factor on bone metabolism and fracture risk (Table 1, Table 2 and Table 3) [19,36]. A study concluded that only levothyroxine treatment is associated with a lower BMD in hypothyroid patients, and also showed that TSH levels in these patients were between 0.1 mIU/L to 2.1 IU/L and overtreatment was the only risk of lower BMD (Table 1, Table 2 and Table 3) [20]. We did not find data regarding whether a correctly balanced treatment of hypothyroidism can reverse the negative effects on bone. 

Autoimmune thyroid disease and bone health. Hashimoto disease can have a unique impact on the the osteoprotegerin/receptor activator of nuclear factor kappa-B ligand (OPG/RANKL) system, apart from the hypothyroidism, potentially leading to additional bone loss [21]. The presence of thyroid peroxidase antibodies (TPOAb) can be a marker of increased fracture risk in euthyroid post-menopausal women [37,38]. 

Subclinical hypothyroidism and bone health. Although previous studies have shown that subclinical hypothyroidism has a reduced BMD and an increased risk of fracture, more recent data contradict those findings (Table 1, Table 2 and Table 3) [33,34,35,36,37]. Since 2014, all of the studies, both prospective and retrospective, have shown no influence of subclinical hypothyroidism on either BMD or fracture risk (Table 1, Table 2 and Table 3) [25,26,41,42,43]. We can conclude that recent data do not support any impact of subclinical hypothyroidism on osteoporosis or fragility fractures.

The same results were obtained in the case of subclinical hypothyroidism treatment with levothyroxine. Newer prospective data contradict previous studies and conclude that treatment with levothyroxine does not affect bone health (Table 1, Table 2 and Table 3) [27,28,29,30].

Subclinical hyperthyroidism and bone health. Subclinical hyperthyroidism is considered by most studies to be associated with a lower BMD and with an increased risk of fracture (Table 1, Table 2 and Table 3) [25,40,42,44,45,46]. Only a few studies showed that subclinical hyperthyroidism is not predictive for incidental hip fracture (Table 1, Table 2 and Table 3) [23,40]. From above results, subclinical hyperthyroidism may be associated with a higher risk of fractures compared to euthyroid patients.

We found only one study to test the impact of radioiodine therapy in the case of subclinical hyperthyroidism (Table 1, Table 2 and Table 3) [29]. It concluded that radioiodine has beneficial effects on BMD, but we believe that more data are needed in order to support the conclusion (Table 1, Table 2 and Table 3) [31].

## 6. Conclusions

Osteoporosis and fragility fractures are important complications of thyroid disorders, associated with increased mortality. Thyroid disorders have an important impact on bone metabolism and fracture risk, as hyperthyroidism and subclinical hyperthyroidism are associated with a decreased BMD and increased risk of fracture. Hypothyroidism overtreatment with levothyroxine has a similar impact on bone health as hyperthyroidism. Subclinical hypothyroidism, on the other hand, is not associated with osteoporosis or fragility fractures and subclinical hyperthyroidism treatment with radioiodine could improve bone health.

## Figures and Tables

**Figure 1 diagnostics-10-00149-f001:**
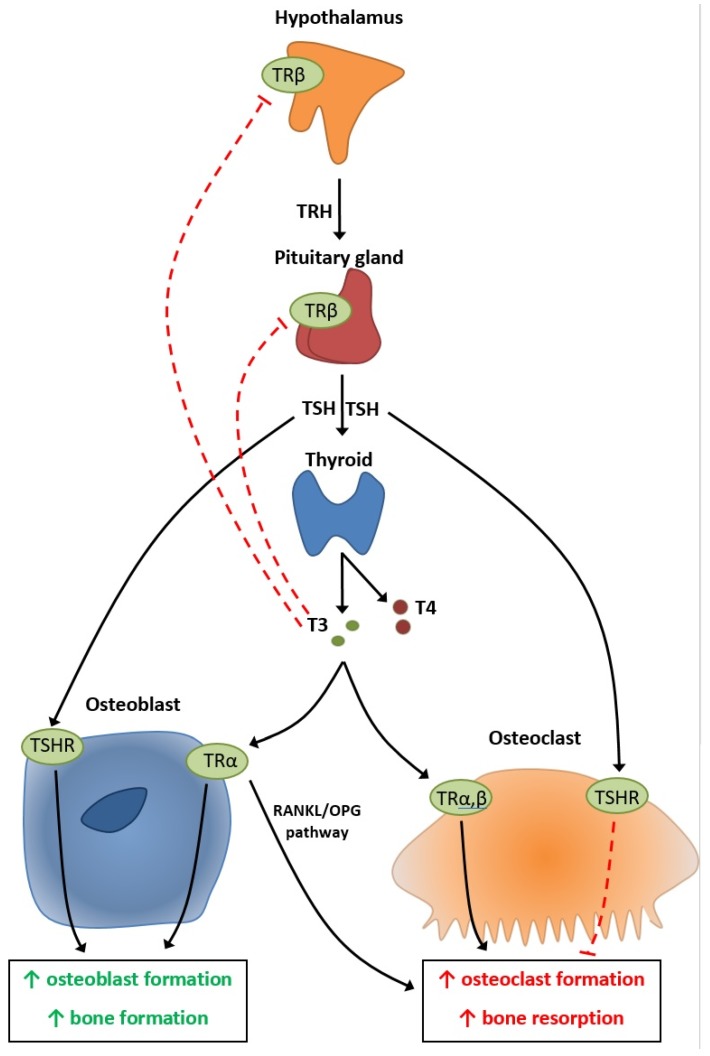
Physiopathology of the hypothalamic–pituitary–thyroid axis on bone metabolism. OPG = osteoprotegenrin; RANKL = receptor activator of nuclear factor kappa-B ligand; TRH = thyrotropin-releasing hormone; TSH = thyroid-stimulating hormone; TSHR = thyroid-stimulating hormone receptor.

**Table 1 diagnostics-10-00149-t001:** Studies regarding the impact of hypothyroidism, autoimmune thyroid pathology, subclinical hypothyroidism, subclinical hyperthyroidism, and their treatments on bone mineral density. BMD = bone mineral density; OPG = osteoprotegenrin; RANKL = receptor activator of nuclear factor kappa-B ligand; TSH = thyroid-stimulating hormone.

Type of Study	Number of Patients	Conclusion	Year	References
**Hypothyroidism**				
Prospective	188	Hypothyroidism is associated with lower BMD	2018	[13]
Cross-sectional	1097	Lower TSH is related to decreased BMD	2016	[14]
Prospective	119	Hypothyroidism does not affect bone density	2013	[15]
Cross-sectional	6722	Lower TSH was related to the lowest BMD in the forearm	2009	[16]
Cross-sectional	959	Lower TSH was related to lower BMD	2006	[17]
**Hypothyroidism treatment**				
Prospective	75	Levothyroxine increases bone turnover, but it does not lead to frank osteoporosis	2015	[18]
Retrospective	11,155	Overtreatment of hypothyroidism is associated with an increased rate of osteoporosis	2014	[19]
Retrospective	150	Levothyroxine is associated with a lower BMD	2014	[20]
**Autoimmune thyroid pathology**				
Prospective	80	Hashimoto disease can further activate the OPG/RANKL system leading to additional bone loss	2016	[21]
**Subclinical hypothyroidism**				
Retrospective	25,510	Subclinical hypothyroidism does not affect BMD	2019	[22]
Prospective	4248	Subclinical hypothyroidism is not predictive of incident hip fracture	2018	[23]
Cross-sectional	1290	TSH does not affect BMD	2012	[24]
Prospective	413	Subclinical hypothyroidism is associated with lower BMD	2006	[25]
Prospective	32	Subclinical hypothyroidism was related to a reduced leg BMD	2002	[26]
**Subclinical hypothyroidism treatment**				
Prospective	196	Levothyroxine does not affect bone health	2020	[27]
Prospective	45	Levothyroxine treatment does not affect bone mass.	2015	[28]
Retrospective	182	Levothyroxine is associated with a higher bone loss	2011	[29]
Cross-sectional retrospective	66	Levothyroxine treatment is related to accelerated bone loss	2004	[30]
**Subclinical hyperthyroidism**				
Prospective	413	Subclinical hyperthyroidism is associated with lower BMD	2006	[25]
**Subclinical hyperthyroidism treatment**				
Prospective	36	Radioiodine has beneficial effects on BMD	2013	[31]

**Table 2 diagnostics-10-00149-t002:** Studies regarding the impact of hypothyroidism, autoimmune thyroid pathology, subclinical hypothyroidism, subclinical hyperthyroidism, and their treatments on fracture risk. TPOAb = thyroid autoantibodies.

Type of Study	Number of Patients	Conclusion	Year	References
**Hypothyroidism**				
Retrospective	162,369	TSH above the upper reference value increases the risk of fractures	2019	[32]
Retrospective	914	TSH above the median increases the early mortality rate in the case of surgically treated hip fractures	2019	[33]
Retrospective	108,977	Hypothyroidism is associated with increased risk of fracture	2016	[34]
Retrospective	16,249	Hypothyroidism is associated with increased fracture risk	2002	[35]
**Hypothyroidism treatment**				
Retrospective	230,552	Overtreatment of hypothyroidism has a similar fracture risk, as seen in hyperthyroidism	2015	[36]
**Autoimmune thyroid pathology**				
Prospective	189	TPOAb can be considered a marker for increased risk of fracture in euthyroid post-menopausal women	2017	[37]
Retroscpective	335	TPOAb can be considered a marker for increased risk of fracture	2017	[38]
**Subclinical hypothyroidism**				
Prospective	4248	Subclinical hypothyroidism is not predictive of incident hip fracture	2018	[23]
Retrospective	471	Subclinical hypothyroidism is not associated with lower BMD	2014	[39]
Cross-sectional	4963	Subclinical hypothyroidism is not associated with increased hip fracture rate	2014	[40]
Prospective	82	Subclinical hypothyroidism has an increased fracture risk	2013	[41]
Prospective	3567	Subclinical hypothyroidism has an increased risk of hip fracture	2010	[42]
**Subclinical hyperthyroidism**				
Prospective	4248	Subclinical hyperthyroidism is not predictive of incident hip fracture	2018	[23]
Cross-sectional	4963	Subclinical hyperthyroidism is not associated with increased hip fracture rate	2014	[40]
Prospective	3567	Subclinical hyperthyroidism has an increased risk of hip fracture	2010	[42]

**Table 3 diagnostics-10-00149-t003:** Meta-analyses regarding the impact of subclinical hypothyroidism and subclinical hyperthyroidism on bone health.

Bone-Related Outcome	Number of Patients	Conclusion	Year	References
**Subclinical hypothyroidism**				
Osteoporosis Fracture risk	313,557	Subclinical hypothyroidism has no impact on fracture risk.	2019	[43]
Fracture risk	314,146	Subclinical hypothyroidism is not associated with an increased risk of fracture.	2016	[44]
Fracture risk	70,298	Subclinical hypothyroidism is not associated with an increased risk of fractures.	2015	[45]
**Subclinical hyperthyroidism**				
Osteoporosis Fracture risk	313,557	Subclinical hyperthyroidism could increase the risk of fractures and decrease BMD.	2019	[40]
Osteoporosis	5458	Subclinical hyperthyroidism is associated with lower BMD.	2018	[46]
Fracture risk	314,146	Subclinical hyperthyroidism is associated with an increased risk of fracture.	2016	[44]
Fracture risk	70,298	Subclinical hyperthyroidism is associated with an increased risk of fractures.	2015	[45]

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
