# Peer review of "The Influence of Thyroid Pathology on Osteoporosis and Fracture Risk: A Review"

_diagnostics, 2020, doi:10.3390/diagnostics10030149_

Round 1
Reviewer 1 Report
The results were adequately commented. I think the result would be more complete if there were also data on autoimmune thyroid disease in subclinical or overt thyroid dysfunction.

Author Response
Dear Reviewer 1,
Thank you for your valuable feedback; the manuscript has been systematically improved, per your advice. We are grateful for your effort and for the relevant observations.
Our answers and required changes are listed below, as follows:
Issue 1: Line 130 – „2.5 SD or less”
Response: The sentece containing this phrase was removed and the paragraph restructured, as suggested by Reviewer 3.
Issue 2: Line 138 – Physiopathology of thyroid’s effect on bone metabolism – thyroid hormone synthesis.
Response: The paragraph regarding thyroid hormone synthesis was remove, as suggested:
“Physiopathology of thyroid’s effect on bone metabolism
The synthesis and the release of the triiodothyronine (T3) and thyroxine (T4) within the thyroid gland are controlled by the hypothalamic-pituitary-thyroid axis [13]. The thyroid releasing hormone (TRH) is released from the hypothalamus and stimulates the secretion of the thyroid-stimulating hormone (TSH) from the pituitary gland (Figure 1) [13]. TSH acts on the thyroid gland and stimulates the production and release of T3 and T4 (Figure 1). This process is regulated by negative feedback, as T3 acts on TRβ receptors on both the hypothalamus and the pituitary gland (Figure 1) [13]. “
Issue 3: Autoimmune thyroid pathology.
Response: We have added data regarding autoimmune thyroid pathology regarding bone impact.
“Materials and methods
We have searched PubMed database for the following keywords: “hypothyroidism osteoporosis”, “hypothyroidism fracture”, “hypothyroidism bone”, “subclinical hypothyroidism osteoporosis”, “subclinical hypothyroidism fracture”, “subclinical hypothyroidism bone”, “subclinical hyperthyroidism osteoporosis”, “subclinical hyperthyroidism fracture”, “subclinical hyperthyroidism bone”, ‘’autoimmune thyroid bone metabolism”, ‘’autoimmune thyroid bone mineral density”, ‘’autoimmune thyroid osteoporosis”, ‘’autoimmune thyroid fracture”, ‘Hashimoto osteoporosis”, ‘’Hashimoto fracture”.”
“Autoimmune thyroid disease and bone health. Hashimoto disease can have a unique impact on the OPG/RANKL system, apart from the hypothyroidism, potentially leading to additional bone loss [30]. The presence of thyroid peroxidase antibodies (TPOAb) can be a marker of increased fracture risk in euthyroid postmenopausal women [31,32]. “
Table 1. Studies regarding the impact of hypothyroidism, autoimmune thyroid pathology, subclinical hypothyroidism, subclinical hyperthyroidism and their treatments on bone mineral density.
|
Type of study |
Number of patients |
Conclusion |
Year |
References |
|
Hypothyroidism |
||||
|
Prospective |
188 |
Hypothyroidism is associated with lower BMD. |
2018 |
[50] |
|
Cross-sectional |
1097 |
Lower TSH is related to decreased BMD. |
2016 |
[21] |
|
Prospective |
119 |
Hypothyroidism does not affect bone density. |
2013 |
[24] |
|
Cross-sectional |
6722 |
Lower TSH was related to the lowest BMD in the forearm |
2009 |
[20] |
|
Cross-sectional |
959 |
Lower TSH was related to lower BMD |
2006 |
[19] |
|
Hypothyroidism - treatment |
||||
|
Prospective |
75 |
Levothyroxine increases bone turnover, but it does not lead to frank osteoporosis. |
2015 |
[51] |
|
Retrospective |
11,155 |
Overtreatment of hypothyroidism is associated with an increased rate of osteoporosis. |
2014 |
[28] |
|
Retrospective |
150 |
Levothyroxine is associated with a lower BMD. |
2014 |
[29] |
|
Autoimmune thyroid pathology |
||||
|
Prospective |
80 |
Hashimoto disease can further activate the OPG/RANKL system leading to additional bone loss. |
2016 |
[30] |
|
Subclinical hypothyroidism |
||||
|
Retrospective |
25,510 |
Subclinical hypothyroidism does not affect BMD. |
2019 |
[38] |
|
Prospective |
4248 |
Subclinical hypothyroidism is not predictive of incident hip fracture. |
2018 |
[39] |
|
Cross-sectional |
1290 |
TSH does not affect BMD. |
2012 |
[52] |
|
Prospective |
413 |
Subclinical hypothyroidism is associated with lower BMD. |
2006 |
[33] |
|
Prospective |
32 |
Subclinical hypothyroidism was related to a reduced leg BMD. |
2002 |
[34] |
|
Subclinical hypothyroidism - treatment |
||||
|
Prospective |
196 |
Levothyroxine does not affect bone health. |
2020 |
[44] |
|
Prospective |
45 |
Levothyroxine treatment does not affect bone mass. |
2015 |
[45] |
|
Retrospective |
182 |
Levothyroxine is associated with a higher bone loss. |
2011 |
[46] |
|
Cross-sectional retrospective |
66 |
Levothyroxine treatment is related to accelerated bone loss. |
2004 |
[47] |
|
Subclinical hyperthyroidism |
||||
|
Prospective |
413 |
Subclinical hyperthyroidism is associated with lower BMD. |
2006 |
[33] |
|
Subclinical hyperthyroidism - treatment |
||||
|
Prospective |
36 |
Radioiodine has beneficial effects on BMD |
2013 |
[49] |
Table 2. Studies regarding the impact of hypothyroidism, autoimmune thyroid pathology, subclinical hypothyroidism, subclinical hyperthyroidism and their treatments on fracture risk.
|
Type of study |
Number of patients |
Conclusion |
Year |
References |
|
Hypothyroidism |
||||
|
Retrospective |
162,369 |
TSH above the upper reference value increases the risk of fractures. |
2019 |
[23] |
|
Retrospective |
914 |
TSH above the median increases the early mortality rate in the case of surgically treated hip fractures. |
2019 |
[22] |
|
Retrospective |
108,977 |
Hypothyroidism is associated with increased risk of fracture. |
2016 |
[25] |
|
Retrospective |
16,249 |
Hypothyroidism is associated with increased fracture risk |
2002 |
[26] |
|
Hypothyroidism - treatment |
||||
|
Retrospective |
230,552 |
Overtreatment of hypothyroidism has a similar fracture risk, as seen in hyperthyroidism. |
2015 |
[27] |
|
Autoimmune thyroid pathology |
||||
|
Prospective |
189 |
TPOAb can be considered a marker for increased risk of fracture in euthyroid postmenopausal women. |
2017 |
[31] |
|
Retroscpective |
335 |
TPOAb can be considered a marker for increased risk of fracture. |
2017 |
[32] |
|
Subclinical hypothyroidism |
||||
|
Prospective |
4248 |
Subclinical hypothyroidism is not predictive of incident hip fracture. |
2018 |
[39] |
|
Retrospective |
471 |
Subclinical hypothyroidism is not associated with lower BMD. |
2014 |
[42] |
|
Cross-sectional |
4963 |
Subclinical hypothyroidism is not associated with increased hip fracture rate. |
2014 |
[43] |
|
Prospective |
82 |
Subclinical hypothyroidism has an increased fracture risk. |
2013 |
[36] |
|
Prospective |
3567 |
Subclinical hypothyroidism has an increased risk of hip fracture. |
2010 |
[35] |
|
Subclinical hyperthyroidism |
||||
|
Prospective |
4248 |
Subclinical hyperthyroidism is not predictive of incident hip fracture. |
2018 |
[39] |
|
Cross-sectional |
4963 |
Subclinical hyperthyroidism is not associated with increased hip fracture rate |
2014 |
[43] |
|
Prospective |
3567 |
Subclinical hyperthyroidism has an increased risk of hip fracture. |
2010 |
[35] |
Table 3. Metaanalysis regarding the impact of subclinical hypothyroidism and subclinical hyperthyroidism on bone health.
|
Bone-related outcome |
Number of patients |
Conclusion |
Year |
References |
|
Subclinical hypothyroidism |
||||
|
Osteoporosis Fracture risk |
313,557 |
Subclinical hypothyroidism has no impact on fracture risk. |
2019 |
[37] |
|
Fracture risk |
314,146 |
Subclinical hypothyroidism is not associated with an increased risk of fracture. |
2016 |
[40] |
|
Fracture risk |
70,298 |
Subclinical hypothyroidism is not associated with an increased risk of fractures. |
2015 |
[41] |
|
Subclinical hyperthyroidism |
||||
|
Osteoporosis Fracture risk |
313,557 |
Subclinical hyperthyroidism could increase the risk of fractures and decrease BMD. |
2019 |
[37] |
|
Osteoporosis |
5458 |
Subclinical hyperthyroidism is associated with lower BMD. |
2018 |
[48] |
|
Fracture risk |
314,146 |
Subclinical hyperthyroidism is associated with an increased risk of fracture. |
2016 |
[40] |
|
Fracture risk |
70,298 |
Subclinical hyperthyroidism is associated with an increased risk of fractures. |
2015 |
[41] |
Reviewer 2 Report
I attached the file.

Author Response
Dear Reviewer 2,
Thank you for your valuable feedback; the manuscript has been systematically improved, per your advice. We are grateful for your effort and for the relevant observations.
Our answers and required changes (marked in red font) are listed below, as follows:
Issue 1: Line 58 – „Primary hypothyroidism causes”
Response: The sentece containing this phrase was removed and the paragraph restructured, as suggested by Reviewer 3.
“The causes of hypothyroidism can be divided into primary, central and peripheral [1]. The primary hypothyroidism is represented by thyroid hormone deficiency and include chronic autoimmune thyroiditis, iodine deficiency or radioiodine treatment; the central hypothyroidism is represented by thyroid stimulating hormone (TSH) deficiency or by thyrotropin-releasing hormone (TRH) deficiency and include pituitary tumors, pituitary dysfunction or hypothalamic dysfunction; while the peripheral hypothyroidism is represented by the peripheral resistance to the thyroid hormones and include decreased sensitivity to thyroid hormone and consumptive hypothyroidism [1].”
Issue 2: Line 97 – „iatrogenic”
Response: The change was done as suggested:
„It can be caused by Graves’ disease, multinodular toxic goiter or autonomous functioning thyroid adenoma, iatrogenic condition in differentiated thyroid carcinoma [3–5].”
Issue 3: Line 111 – „different”
Response: The change was done as suggested:
„Secondary causes are numerous and they include liver disease, renal failure, rheumatoid arthritis, chronic obstructive pulmonary disease, endocrine disorders, gastrointestinal disorders, hematologic disorders, nutrition disorders, central nervous system disorders, human immunodeficiency virus infection, systemic lupus erythematosus and diverse medications [8–10].
Issue 4: Line 114 – „if”
Response: The change was done as suggested:
“as it increases the fracture risk by two-fold for every standard deviation below the mean of a young adult [7,11]. “
Issue 5: Line 123 – „vitamin D is needed to be associated”
Response: The change was done as suggested:
„Moreover, vitamin D intake is also needed at a dose of 600 IU/day until 70 years old and 800 IU/day afterward [6].”
Issue 6: Line 127 – „begin”
Response: The change was done as suggested:
“Other management strategies for osteoporotic patients in order to reduce the fracture risk include the decrease of alcohol consumption, less than two servings of caffeine drinks daily, beginning a regular physical exercise program, fall-prevention programs.”
Issue 7: Line 148 – „osteoclast”
Response: The change was done as suggested:
„They are connected to each other by a dendritic network and initiate the bone resorption performed by osteoclasts”
Issue 8: Line 154 – „decrease”
Response: The change was done as suggested:
„On osteoblasts it is considered to increase osteoblast formation and bone formation, while on the osteoclasts it could increase osteoclast formation and the bone resorption process (Figure 1) [13].”
Issue 9: Line 186 – „but only”
Response: The change was done as suggested:
„Only one prospective study performed by Tuchendler and Bolanowski in 2013 on 119 participants did not show a negative effect on BMD in the case of hypothyroidism, except in the case of hyperthyroidism (Tables 1,2,3) [24].”
Issue 10: Line 187 – „108,977”
Response: The change was done as suggested:
„Regarding the incidence of fracture risk in hypothyroidism patients, we found two studies on 92,341 and 16,249 patients,”
Issue 11: Line 188 – „an increased risk of”
Response: The change was done as suggested:
„which showed a higher incidence of fractures in these patients (Tables 1,2,3) [25,26].”
Issue 12: Line 190 – „Can the treatment of hypothyroidism reverse the negative effect on bone?”
Response: The change was done as suggested:
„We did not find data regarding whether a correctly balanced treatment of hypothyroidism can reverse the negative effect on bone.”
Issue 13: Lines 193-195– „A study that concluded that levothyroxine treatment alone in the case of hypothyroidism is associated with a lower BMD (Table 1) [29]. Nevertheless, in this study, the TSH levels in the hypothyroidism patients were between 0.1 to 2.1 mIU/L, therefore suggesting overtreatment in some of the patients (Table 1) [29].”
Response: The change was done as suggested:
„A study concluded that only levothyroxine treatment is associated with a lower BMD in hypothyroid patients also showed that TSH levels in these patients were between 0.1 to 2.1 1 mIU/L and only overtreatment is the risk of lower BMD (Tables 1,2,3) [29].”

Reviewer 3 Report
The manuscript is a review of an interestly field of actual discussion.
The order of the presentation is not correct. After the introduction and material and methods, it would probable be followed by the section of Bone health and Physiophatology of thryoid´s effect on bone. Then Thyroid pathology
In material & methods the authors should explain the criteria for selecting bone studies (DXA, TBS, peripheral CT...). And how many paper were discharged.
Thyroid pathology. The description of hypothyroidism symptomatology isnot necessary and not further discussed. The same respect the classification This section can be summarized.
Bone health section is well written. Please clearly define that osteoporosis can be asymtomatic, but symtomatic with clinical fractures.Therapy of osteoporosis does not seems to be necessary to discuss in this paper.
The description of bone loss and fractures should be separate.For example table1, is confusing. References 22 refere to mortality after hip fracture, ref. 25 & 26 with increase fracture risk (vertebral ??=). Also in table meta-analisis refer. are included, that also includes the other papers. I suggest to order the publications with bone mineral density loss by years, and then the same for fracture. Meta-analysis should be discuss separately-
Author Response
Dear Reviewer 3,
Thank you for your valuable feedback; the manuscript has been systematically improved, per your advice. We are grateful for your effort and for the relevant observations.
Our answers and required changes are listed below, as follows:
Issue 1: The order of the presentation is not correct. After the introduction and material and methods, it would probable be followed by the section of Bone health and Physiophatology of thryoid´s effect on bone. Then Thyroid pathology.
Response: We have changed the order of the chapters accordingly.
Issue 2: In material & methods the authors should explain the criteria for selecting bone studies (DXA, TBS, peripheral CT...). And how many paper were discharged.
Response: We have explained the criteria for the selection of bone studies and the number of papers which were excluded. The changes are found below:
„We have included clinical trials and meta-analysis since 2002 until 2020. The bone studies reviewed in our study included DXA scan, bone turnover markers and FRAX score. From a total of 3505 articles, only 41 fulfilled our inclusion criteria.”
Issue 3: Thyroid pathology. The description of hypothyroidism symptomatology is not necessary and not further discussed. The same respect the classification This section can be summarized.
Response: We have made the required changes in the thyroid pathology chapter, as following:
“Hypothyroidism is represented by thyroid hormone deficiency and it is a common pathology worldwide, with a prevalence of up to 7% in the general population [1]. The causes of hypothyroidism can be divided into primary, central and peripheral [1]. The primary hypothyroidism is represented by thyroid hormone deficiency and include chronic autoimmune thyroiditis, iodine deficiency or radioiodine treatment; the central hypothyroidism is represented by thyroid stimulating hormone (TSH) deficiency or by thyrotropin-releasing hormone (TRH) deficiency and include pituitary tumors, pituitary dysfunction or hypothalamic dysfunction; while the peripheral hypothyroidism is represented by the peripheral resistance to the thyroid hormones and include decreased sensitivity to thyroid hormone and consumptive hypothyroidism [1].
Although the hypothyroidism patients can be asymptomatic, the clinical presentation of hypothyroidism can include numerous symptoms such as weight gain, cold intolerance, fatigue, shortness of breath, impaired memory, paresthesia, constipation, infertility, galactorrhea, menstrual disturbance, muscle cramps, muscle weakness, arthralgia, dry skin, hair loss or deterioration of kidney function, amenorrhea and pituitary pathology [1]. Life-threatening manifestations are present in the case of myxedema coma, where the patients develop altered mental status, progressive lethargy, hypothermia and bradycardia [1]. “
Issue 4: Bone health section is well written. Please clearly define that osteoporosis can be asymtomatic, but symtomatic with clinical fractures.Therapy of osteoporosis does not seems to be necessary to discuss in this paper.
Response: We have made the required changes in the bone health chapter, as following:
„Clinically, osteoporosis has no symptoms, but it can lead to loss of height, back pain, change in posture or intense pain following a fracture, as it increases the fracture risk by two-fold for every standard deviation below the mean of a young adult [7,11].“
Issue 5: The description of bone loss and fractures should be separate.For example table1, is confusing. References 22 refere to mortality after hip fracture, ref. 25 & 26 with increase fracture risk (vertebral ??=). Also in table meta-analisis refer. are included, that also includes the other papers. I suggest to order the publications with bone mineral density loss by years, and then the same for fracture. Meta-analysis should be discuss separately-
Response: We have split the data into the three tables, as suggested. The changes are found below.
Table 1. Studies regarding the impact of hypothyroidism, autoimmune thyroid pathology, subclinical hypothyroidism, subclinical hyperthyroidism and their treatments on bone mineral density.
|
Type of study |
Number of patients |
Conclusion |
Year |
References |
|
Hypothyroidism |
||||
|
Prospective |
188 |
Hypothyroidism is associated with lower BMD. |
2018 |
[50] |
|
Cross-sectional |
1097 |
Lower TSH is related to decreased BMD. |
2016 |
[21] |
|
Prospective |
119 |
Hypothyroidism does not affect bone density. |
2013 |
[24] |
|
Cross-sectional |
6722 |
Lower TSH was related to the lowest BMD in the forearm |
2009 |
[20] |
|
Cross-sectional |
959 |
Lower TSH was related to lower BMD |
2006 |
[19] |
|
Hypothyroidism - treatment |
||||
|
Prospective |
75 |
Levothyroxine increases bone turnover, but it does not lead to frank osteoporosis. |
2015 |
[51] |
|
Retrospective |
11,155 |
Overtreatment of hypothyroidism is associated with an increased rate of osteoporosis. |
2014 |
[28] |
|
Retrospective |
150 |
Levothyroxine is associated with a lower BMD. |
2014 |
[29] |
|
Autoimmune thyroid pathology |
||||
|
Prospective |
80 |
Hashimoto disease can further activate the OPG/RANKL system leading to additional bone loss. |
2016 |
[30] |
|
Subclinical hypothyroidism |
||||
|
Retrospective |
25,510 |
Subclinical hypothyroidism does not affect BMD. |
2019 |
[38] |
|
Prospective |
4248 |
Subclinical hypothyroidism is not predictive of incident hip fracture. |
2018 |
[39] |
|
Cross-sectional |
1290 |
TSH does not affect BMD. |
2012 |
[52] |
|
Prospective |
413 |
Subclinical hypothyroidism is associated with lower BMD. |
2006 |
[33] |
|
Prospective |
32 |
Subclinical hypothyroidism was related to a reduced leg BMD. |
2002 |
[34] |
|
Subclinical hypothyroidism - treatment |
||||
|
Prospective |
196 |
Levothyroxine does not affect bone health. |
2020 |
[44] |
|
Prospective |
45 |
Levothyroxine treatment does not affect bone mass. |
2015 |
[45] |
|
Retrospective |
182 |
Levothyroxine is associated with a higher bone loss. |
2011 |
[46] |
|
Cross-sectional retrospective |
66 |
Levothyroxine treatment is related to accelerated bone loss. |
2004 |
[47] |
|
Subclinical hyperthyroidism |
||||
|
Prospective |
413 |
Subclinical hyperthyroidism is associated with lower BMD. |
2006 |
[33] |
|
Subclinical hyperthyroidism - treatment |
||||
|
Prospective |
36 |
Radioiodine has beneficial effects on BMD |
2013 |
[49] |
Table 2. Studies regarding the impact of hypothyroidism, autoimmune thyroid pathology, subclinical hypothyroidism, subclinical hyperthyroidism and their treatments on fracture risk.
|
Type of study |
Number of patients |
Conclusion |
Year |
References |
|
Hypothyroidism |
||||
|
Retrospective |
162,369 |
TSH above the upper reference value increases the risk of fractures. |
2019 |
[23] |
|
Retrospective |
914 |
TSH above the median increases the early mortality rate in the case of surgically treated hip fractures. |
2019 |
[22] |
|
Retrospective |
108,977 |
Hypothyroidism is associated with increased risk of fracture. |
2016 |
[25] |
|
Retrospective |
16,249 |
Hypothyroidism is associated with increased fracture risk |
2002 |
[26] |
|
Hypothyroidism - treatment |
||||
|
Retrospective |
230,552 |
Overtreatment of hypothyroidism has a similar fracture risk, as seen in hyperthyroidism. |
2015 |
[27] |
|
Autoimmune thyroid pathology |
||||
|
Prospective |
189 |
TPOAb can be considered a marker for increased risk of fracture in euthyroid postmenopausal women. |
2017 |
[31] |
|
Retroscpective |
335 |
TPOAb can be considered a marker for increased risk of fracture. |
2017 |
[32] |
|
Subclinical hypothyroidism |
||||
|
Prospective |
4248 |
Subclinical hypothyroidism is not predictive of incident hip fracture. |
2018 |
[39] |
|
Retrospective |
471 |
Subclinical hypothyroidism is not associated with lower BMD. |
2014 |
[42] |
|
Cross-sectional |
4963 |
Subclinical hypothyroidism is not associated with increased hip fracture rate. |
2014 |
[43] |
|
Prospective |
82 |
Subclinical hypothyroidism has an increased fracture risk. |
2013 |
[36] |
|
Prospective |
3567 |
Subclinical hypothyroidism has an increased risk of hip fracture. |
2010 |
[35] |
|
Subclinical hyperthyroidism |
||||
|
Prospective |
4248 |
Subclinical hyperthyroidism is not predictive of incident hip fracture. |
2018 |
[39] |
|
Cross-sectional |
4963 |
Subclinical hyperthyroidism is not associated with increased hip fracture rate |
2014 |
[43] |
|
Prospective |
3567 |
Subclinical hyperthyroidism has an increased risk of hip fracture. |
2010 |
[35] |
Table 3. Metaanalysis regarding the impact of subclinical hypothyroidism and subclinical hyperthyroidism on bone health.
|
Bone-related outcome |
Number of patients |
Conclusion |
Year |
References |
|
Subclinical hypothyroidism |
||||
|
Osteoporosis Fracture risk |
313,557 |
Subclinical hypothyroidism has no impact on fracture risk. |
2019 |
[37] |
|
Fracture risk |
314,146 |
Subclinical hypothyroidism is not associated with an increased risk of fracture. |
2016 |
[40] |
|
Fracture risk |
70,298 |
Subclinical hypothyroidism is not associated with an increased risk of fractures. |
2015 |
[41] |
|
Subclinical hyperthyroidism |
||||
|
Osteoporosis Fracture risk |
313,557 |
Subclinical hyperthyroidism could increase the risk of fractures and decrease BMD. |
2019 |
[37] |
|
Osteoporosis |
5458 |
Subclinical hyperthyroidism is associated with lower BMD. |
2018 |
[48] |
|
Fracture risk |
314,146 |
Subclinical hyperthyroidism is associated with an increased risk of fracture. |
2016 |
[40] |
|
Fracture risk |
70,298 |
Subclinical hyperthyroidism is associated with an increased risk of fractures. |
2015 |
[41] |

Round 2
Reviewer 3 Report
I have carefully review the revised manuscript. The authors have made all the changes that I suggested, and therefore I believe now can be ready for publication in the journal.